# Relax, it doesn't matter how you get there: A new self-supervised approach for multi-timescale behavior analysis

**Mehdi Azabou**[*]
Georgia Tech

**Michael J. Mendelson**
Georgia Tech

**Nauman Ahad**
Georgia Tech

**Maks Sorokin**
Georgia Tech

**Shantanu Thakoor**
DeepMind

**Carolina Urzay**
Georgia Tech

**Eva L. Dyer**[*]
Georgia Tech

## Abstract

Unconstrained and natural behavior consists of dynamics that are complex and unpredictable, especially when trying to predict what will happen multiple steps into the future. While some success has been found in building representations of animal behavior under constrained or simplified task-based conditions, many of these models cannot be applied to free and naturalistic settings where behavior becomes increasingly hard to model. In this work, we develop a multi-task representation learning model for animal behavior that combines two novel components: (i) an action-prediction objective that aims to predict the distribution of actions over future timesteps, and (ii) a multi-scale architecture that builds separate latent spaces to accommodate short- and long-term dynamics. After demonstrating the ability of the method to build representations of both local and global dynamics in robots in varying environments and terrains, we apply our method to the MABe 2022 Multi-Agent Behavior challenge, where our model ranks first overall on both mice and fly benchmarks. In all of these cases, we show that our model can build representations that capture the many different factors that drive behavior and solve a wide range of downstream tasks.

## 1 Introduction

Behavior is shaped by various factors operating across different timescales. Immediate motivations can drive moment-to-moment interactions, while long-term experiences or even the time of day can influence behavior on broader scales. Analyzing these dynamics, particularly in complex and naturalistic contexts (34; 28), has now become a critical component in many modern studies in neuroscience (28), cognitive science, and in social behavior and decision making (38; 52; 56; 5). Additionally, monitoring and tracking systems now allow for modeling of multi-agent interactions (27; 11; 16) and social behaviors (52; 1), providing valuable insights into dynamics across many individuals.

In order to learn latent factors that may influence behavioral patterns, a promising solution is to build models of behavior in a unsupervised manner (25; 68). Unsupervised models are of particular interest in this domain as it becomes hard to identify complex behaviors which can be composed of many "syllables" of movement (65), and are thus hard and tedious to annotate. Recent work in this direction build such representations using generative modeling and reconstruction-based objectives,

---

[*]Contact: {mazabou,evadyer}@gatech.edu. Project page and code: https://multiscale-behavior.github.io/

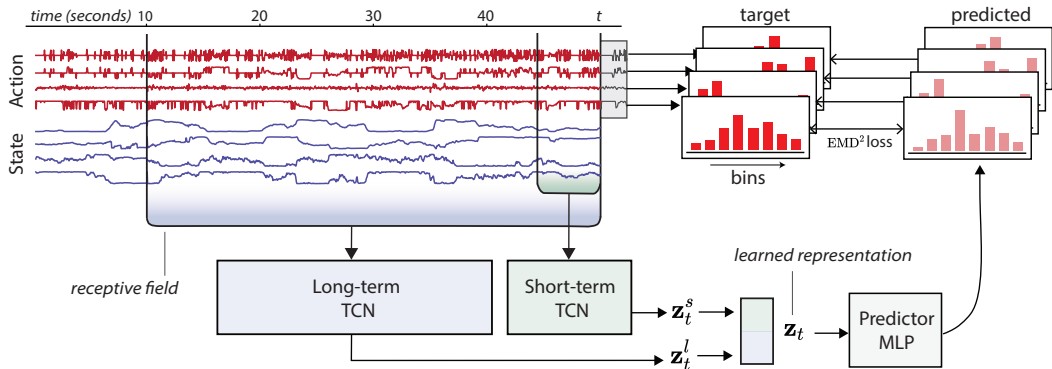

Figure 1: *Overview of our approach*. Bootstrap Across Multiple Scales (BAMS) uses two temporal convolutional networks with two latent spaces, each with their own receptive field sizes. The model is trained on a novel learning objective that consists in predicting future action distributions instead of future action sequences. In this figure, we use sample data from the MABe dataset, only a subset of the channels are shown.

typically by performing open loop (58; 13; 10) or closed loop (13) prediction of observations or actions multiple timesteps into the future.

However, when using a reconstruction or prediction objective to analyze behavior, future actions become hard to predict and models can start to become myopic, focusing only on short-range interactions in the data (65). To circumvent this overly local learning of dynamics, there have been a number of efforts to build models of long-term behavioral style (41; 6), where instance-level learning methods are used to extract a single representation for an entire sequence. However, these models then lose their ability to provide time-varying representations that capture the dynamic nature of different behaviors. It is still an outstanding challenge to build representations that can capture both short-term behavioral dynamics along with longer-term trends and global structure.

In this work, we develop a new self-supervised approach for learning multiscale representations of behavior. Our method consists of two core innovations: (i) a novel action-prediction approach that aims to predict the *distribution of actions* over future timesteps, without modeling exactly when each action is taken, and (ii) a novel multi-scale architecture that builds *separate latent spaces to accommodate short- and long-term dynamics*. We combine both of these innovations and show that our approach can capture both long-term and short-term attributes of behavior and work flexibly to solve a variety of different downstream tasks.

To test our approach, we utilize behavioral datasets that contain multiple tasks that vary in complexity and contain distinct multi-timescale dynamics. Using NVIDIA's Isaac Gym (37), we generate a synthetic dataset of the multi-limb kinematics from quadruped robots by varying the robot's morphological properties and the environment's terrain type and difficulty. Using this robot behavior data, we demonstrate that our method can effectively build dynamical models of behavior that accurately elicit both the robot and environment properties, without any explicit training signal encouraging the learning of this information.

Having established that our approach can successfully predict the complex behavior of an artificial creature, we apply it to two multi-agent behavior benchmarks (61) and challenges with multiple tasks that vary in their frame-level (local) vs. sequence-level (global) labels and properties. On the mouse triplet benchmark, we rank first overall on the leaderboard [1] (averaged across 13 tasks), first on all of the 4 global tasks, and are in the top-3 on all the 9 frame-level local subtasks. In one of the global tasks (decoding the strain of the mouse), we achieve impressive performance over the other methods, with a 10% gap over the next best performing method. On the fruit fly groups benchmark, we also rank first overall [2] (averaged across 50 tasks), and outperform other methods on both average frame-level and sequence-level subtasks. Our results demonstrate that our approach can provide representations that can be used to decode meaningful information from behavior that spans many timescales (longer approach interactions, grooming, etc.) as well as global attributes like the time of day or the strain of the mouse.

---

[1] aicrowd.com/challenges/multi-agent-behavior-challenge-2022/problems/mabe-2022-mouse-triplets/leaderboards?post_challenge=true

[2] aicrowd.com/challenges/multi-agent-behavior-challenge-2022/problems/mabe-2022-fruit-fly-groups/leaderboards?post_challenge=true

The contributions of this work include:

- *A self-supervised framework that learns representations in two separate long-term and short-term embedding spaces.* Bootstrapping is performed within each timescale, using a latent predictive loss across positive views only, and hence can process much longer sequences than contrastive methods that require negative views.

- *A novel prediction task for behavioral analysis and cloning* called HoA (histogram of actions), which aims to predict the future distribution of keypoints instead of the precise ordering of future states. We use an efficient implementation of the 2-Wasserstein divergence as a measure of distributional fit between the true movement distribution and the predicted movements.

- *New state-of-the-art performance*: When applied to the Multi-agent (MABe) Benchmark, our model is ranked #1 overall and achieves top scores on all of the sequence-level (global) tasks. In aggregate, we achieve the top rank on the leaderboard.

- *A procedurally generated dataset of walking quadruped robots* with ground truth annotations and multiple frame- and sequence-level tasks. We believe that this dataset can provide a robust benchmark for future work in this direction.

## 2 Method

Our approach addresses two critical challenges of modeling naturalistic behavior (Figure 1). In Section 2.2, we introduce a novel distributional-relaxation of the reconstruction-based learning objective. In Section 2.3, we introduce an architecture and self-supervised learning objectives that support the learning of behavior at different levels of temporal granularity.

### 2.1 Problem setup

We assume a fixed dataset of $D$ trajectories, each comprised of a sequence of observations $\mathbf{x}_t$ and/or actions $\mathbf{y}_t$. Where actions are not explicitly provided, in many cases we can infer actions based on the difference between consecutive observations. Our goal is to learn, for each timestep, behavioral representations $\mathbf{z}_t$ that capture both global-information such as the strain of the mouse or the time of day, as well as temporally-localised representations such as the activity each mouse is engaged in at a given point in time. As obtaining labeled datasets for realistically-useful scales of agent population and diversity of behavior is impractical, we aim to learn representations in an unsupervised manner.

### 2.2 Histogram of Actions (HoA): A novel objective for predicting future

Modeling behavior dynamics can be done by training a model to predict future actions. This reconstruction-based objective becomes challenging when behavior is complex and non-stereotyped. Let us consider the example of a mouse scanning the room, rotating its head from one side to the other. It is possible to extrapolate the trajectory of the head over a few milliseconds, but prediction quickly becomes impossible, not because this particular behavior is complex but because any temporal misalignment in the prediction leads to increasing errors.

We propose to predict the distribution of future actions rather than their sequence. The motivation behind this distributional-relaxation of the reconstruction objective lies in *blurring the exact temporal unfolding of the actions* while preserving their behavioral fingerprint.

**Predicting histograms of future actions.** Let $\mathbf{y}_t \in \mathbb{R}^N$ be the action vector at time $t$. Each feature in the action vector can, for example, represent the linear velocity of a joint or the angular velocity of the head. Given observations $[\mathbf{x}_0, \ldots, \mathbf{x}_t]$ of the behavior at timesteps 0 through $t$, the objective is to predict the distribution of future actions over the next $L$ timesteps. For each $i$-th element of the action vector, we compute a one-dimensional normalized histogram of the values it takes between timesteps $t+1$ and $t+L$. We pre-partition the space of action values into $K$ equally spaced bins, resulting in a $K$-dimensional histogram that we denotes as $\mathbf{h}_{t,i}$, for all keypoints $1 \leq i \leq N$.

We introduce a predictor $g$ that, given the extracted representation $\mathbf{z}_t$, predicts all feature-wise histograms of future actions. The predictor is a multi-layer perceptron (MLP) with an output space in $\mathbb{R}^{N \times K}$. The output is split into $N$ vectors, which are normalized using the softmax operator. We obtain $[\hat{\mathbf{h}}_{t,1}, \ldots, \hat{\mathbf{h}}_{t,n}]$, each estimating the histogram of the $i$-th action feature following timestep $t$.

**EMD² loss for histograms.** To measure the loss between the predicted and target histograms, we use the Discrete Wasserstein distance, also known as the Earth Mover's Distance (EMD). This distance is obtained by solving an optimal transport problem that consists in moving mass from one distribution to the other while incurring the lowest transport cost. In our case, the cost of moving mass from one bin to another is equal to the number of steps between the two bins.

Because our histogram has equally-sized bins, the EMD is equivalent to the Mallows distance which has a closed-form solution ([31]; [20]). In particular we use EMD² which has been shown to be easier to optimize and converge faster ([53]). The loss is defined as follows:

$$\mathcal{D}_{\mathrm{EMD^2}}(\mathbf{h}_{t,i}, \hat{\mathbf{h}}_{t,i}) = \sum_{k=1}^{K}(\mathrm{CDF}_k(\mathbf{h}_{t,i}) - \mathrm{CDF}_k(\widehat{\mathbf{h}}_{t,i}))^2, \tag{1}$$

where $\mathrm{CDF}_k(\mathbf{h})$ is the $k$-th element of the cumulative distribution function of $\mathbf{h}$.

The total loss is obtained by summing over all features of the action vector, which leaves us with the following loss at time $t$:

$$\mathcal{L}_t = \sum_{i=1}^{N} \mathcal{D}_{\mathrm{EMD^2}}(\mathbf{h}_{t,i}, \widehat{\mathbf{h}}_{t,i}). \tag{2}$$

### 2.3   Multi-timescale bootstrapping in a temporally-diverse architecture

In order to form richer and multi-scale representations of behavior, we use a self-supervised learning objective. We introduce a new approach using latent predictive losses to build representations across different scales while preserving the granularity in each. We achieve this by explicitly separating the short-term and long-term representations, then bootstrapping within each representation space. This approach enables us to learn from otherwise incompatible representation learning objectives ([55]; [62]).

#### 2.3.1   Two latent spaces are better than one

Our goal is to capture and separate short-term and long-term dynamics in two different spaces. We use the Temporal Convolutional Network (TCN) ([4]) as our building block. The TCN produces a representation at time $t$ that only depends on the past observations ([46]).

We design an architecture that separates the different timescales by using two TCN encoders: A **short-term encoder**, $f_s$, that captures short-term dynamics and targets momentary behaviors such as drinking, running or chasing; A **long-term encoder**, $f_l$, that captures long-term dynamics and targets longstanding factors that modulate behavior (strain of mouse, time of day). Architecturally, the difference between the two is that we increase the number of layers and use larger dilation rates ([8]) for the long-term encoder, thus effectively covering a larger receptive field (more history) in the input sequence. All feature embeddings extracted by the TCNs are concatenated, to produce the final embedding, $\mathbf{z}_t = \mathbf{concat}[\mathbf{z}_t^s, \mathbf{z}_t^l]$.

#### 2.3.2   Bootstrapping Across Multiple Scales

We draw inspiration from recent work ([17]; [50]; [18]) that uses latent bootstrapping to learn a latent space where "positive" views are mapped close to each other. Unlike other contrastive methods ([69]), bootstrapping does not require negative examples.

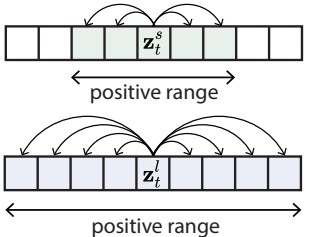

In the context of temporal representation learning, a common assumption is that points that are nearby in time are positive views of each other and can be constrained to lie nearby in the latent space ([69]; [3]). In our case, we can bootstrap and find positive views at both the short-term and also at a more long-term scale, as illustrated in Figure 2.

Figure 2: *Visualization of the short-term and long-term windows used to build multi-scale similarity.* Positives are selected within a window. The window is small for short-term embeddings and can be as large as the entire sequence for long-term embeddings.

**Bootstrapping short-term representations.** We randomly select samples, both future or past, that are within a small window $\Delta$ of the current timestep $t$. In other words, $\delta \in [-\Delta, \Delta]$. Bootstrapping involves using a shallow network to predict the representation of one view from the other ([17]). We

use a predictor $q_s$ that takes in the short-term embedding $\mathbf{z}_t^s$ and learns to regress $\mathbf{z}_{t+\delta}^s$ using the loss:

$$\mathcal{L}_{r,short} = \left\| \frac{q_s(\mathbf{z}_t^s)}{\|q_s(\mathbf{z}_t^s)\|_2} - \text{sg}\left[ \frac{\mathbf{z}_{t+\delta}^s}{\|\mathbf{z}_{t+\delta}^s\|_2} \right] \right\|_2^2, \tag{3}$$

where $\text{sg}[\cdot]$ denotes the stop gradient operator. Unlike (17), we do not use an exponential moving average of the model, but simply increase the learning rate of the predictor as in (50).

**Bootstrapping long-term representations.** For long-term behavior embeddings which should be stable at the level of a sequence, we sample any other time point in the same sequence, i.e. $t' \in [0, T]$. We use a similar setup for the long-term behavior embedding, where predictor $q_l$ is trained over longer time periods or in the limit, over the entire sequence.

$$\mathcal{L}_{r,long} = \left\| \frac{q_l(\mathbf{z}_t^l)}{\|q_l(\mathbf{z}_t^l)\|_2} - \text{sg}\left[ \frac{\mathbf{z}_{t'}^l}{\|\mathbf{z}_{t'}^l\|_2} \right] \right\|_2^2 \tag{4}$$

## 2.4 Putting it all together

Finally, we optimize the proposed multi-task architecture with a combined loss:

$$\mathcal{L} = \mathcal{L}_t + \alpha(\mathcal{L}_{r,short} + \mathcal{L}_{r,long}) \tag{5}$$

where $\alpha$ is a scalar that is used to weigh the contribution of the short- and long-term contrastive losses. In practice, we find that we simply need to choose $\alpha$ that re-scales the contrastive losses to the same order of magnitude as the HoA prediction loss.

# 3 Experiments

## 3.1 Simulated Quadrupeds Experiment

### 3.1.1 A synthetic dataset of simulated legged robots

To test our model's ability to separate behavioral factors that vary in complexity and contain distinct multi-timescale dynamics, we introduce a new dataset generated from a heterogeneous population of quadrupeds traversing different terrains. Simulation enables access to information that is generally inaccessible or hard to acquire in a real-world setting and provides ground-truth information about the agent and the world state.

**Agents.** We use advanced robotic systems (33) that imitates 4-legged creatures capable of various locomotion skills. These robots are trained to walk on challenging terrains using reinforcement learning (51). We use two robots that differ by their morphology, ANYmal B and ANYmal C. To create heterogeneity in the population, we randomize the body mass of the robot as well as the target traversal velocity. We track a set of 24 proprioceptive features including linear and angular velocities of the robots' joints.

**Procedurally generated environments.** Using NVIDIA's Isaac Gym (33) simulation environment, we procedurally generate maps composed of multiple segments of different terrains types (Figure 3). We consider five different terrains including flat surfaces, pits, hills, ascending and descending stairs. We also vary the roughness and slope of the terrain to control the difficulty of terrain traversal.

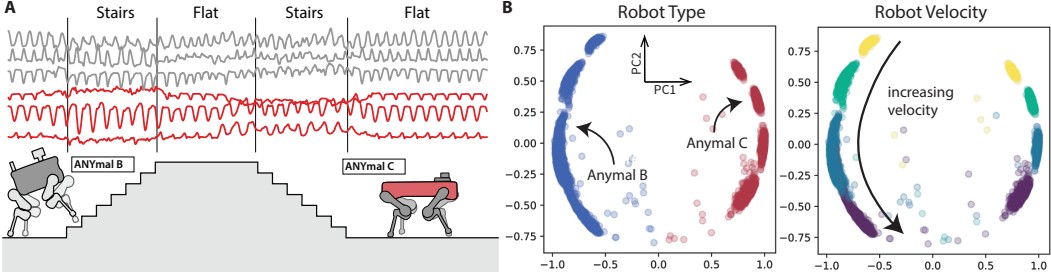

Figure 3: *Quadrupeds walking on procedurally generated map.* (A) Illustration showing the two robots (ANYmal B and ANYmal C) walking on the procedurally generated map, with segments of different terrain types. As the robots traverse different terrains, the velocity of their joints is tracked and visualized. (B) Long-term embeddings learned by BAMS. On the left, we overlay the labels for the type of robot and on the right we overlay the velocity of the robot; we observe a clear organization of the latents in terms of their velocity and robot type.

| | Sequence-level Tasks | | Frame-level Tasks | | |
|---|---|---|---|---|---|
| Model | Robot Type* (↑) | Linear velocity (↓) | Terrain type* (↑) | Terrain slope (↓) | Terrain difficulty (↓) |
| PCA | 99.72 | 0.069 | 08.83 | 0.037 | 0.790 |
| TCN | 99.93 | 0.102 | 33.03 | 0.037 | 0.080 |
| BAMS | 99.96 | 0.038 | **39.89** | **0.033** | **0.078** |
| ↳ short-term | **100.0** | 0.094 | 34.86 | 0.036 | 0.079 |
| ↳ long-term | 99.88 | **0.020** | 32.39 | 0.036 | **0.078** |

Table 1: *Linear readouts of robot behavior.* For each task, we report the linear decoding performance on sequence-level and frame-level tasks. Tasks marked with * are classification tasks for which the F1-score is reported, for the remaining tasks, we report the mean-squared error.

**Experimental setup.** We collect 5182 trajectories of robots walking through terrains. We record for 3 minutes at a frequency of 50Hz. For evaluation, we split the dataset into train and test sets (80/20 split) and use multi-task probes that correspond to different long-term and short-term behavioral factors. More details can be found in Appendix A.

### 3.1.2 Results

Results in Table 1 suggest that our model performs well on these diverse prediction tasks. A major advantage of our method is the separation of the short-term and long-term dynamics, which enables us to more clearly identify the multi-scale factors. While some of the tasks are represented best in the mixed model, we find that the linear velocity is more decodable in the long-term embedding, while terrain type is more decodable in the short-term embedding. We further analyze the formed representations by visualizing the embeddings in the different spaces. In Figure 3-B, we visualize the long-term embedding space and find that our model is able to capture the main factors of variance in the dataset, corresponding to the robot type and the velocity at which the robots are moving. This suggest that in the absence of labels, the learned embedding can provide valuable insights into how the recorded population is distributed without the need for annotations. In Appendix A, we visualize the extracted embeddings of a single sequence over time. We find that the long-term embedding is more stable and smooth, while the short term embeddings reveal different blocks of behavior that change more frequently.

## 3.2 Experiments on Mouse Triplet Dataset

### 3.2.1 Experimental Setup and Tasks

**Dataset description.** The mouse triplet dataset (61) is part of the Multi-Agent Behavior Challenge (MABe 2022), hosted at CVPR 2022. This large-scale dataset was introduced to address the lack of standardized benchmarks for representation learning of animal behavior. It consists of a set of trajectories from three mice interacting in an open-field arena. A total of 5336 one-minute clips, recorded from a top-view camera, were curated and processed to track twelve anatomically defined keypoints on each mouse, as shown in Figure 4.

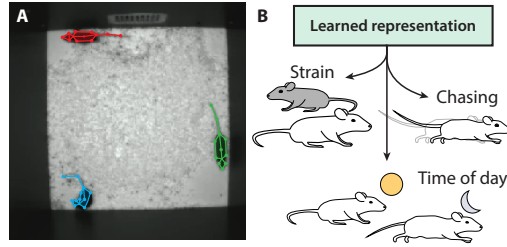

Figure 4: *Multi-Agent Behavior (MABe) - Mouse Triplets Challenge.* (A) Keypoint tracking approaches are used to extract keypoints from many positions on the mouse body in a video. (B) Methods are evaluated across 13 different tasks.

As part of this benchmark, a set of 13 common behavior analysis tasks were identified, and are used to evaluate the performance of representation learning methods. Over the course of these sequences, the mice might exhibit individual and social behaviors. Some might unfold at the frame level, like chasing or being chased, others at the sequence level, like light cycles affecting the behavior of the mice or mouse strains that inherently differentiate behavior.

**Integrating features across multiple animals.** We process the trajectory data to extract 36 features characterizing each mouse individually, including head orientation, body velocity and joint angles. We construct the short-term TCN and long-term TCN encoders to have representation of size 32 each, and receptive fields approximated to be 60 and 1200 frames respectively. We compute the histogram of actions over 1 second, i.e. $L = 30$ and use $K = 32$ bins. We build representations for each mouse independently, which means that at time t, and for each frame $t$, we produce embeddings $\mathbf{z}_{t,1}$, $\mathbf{z}_{t,2}$

| | Sequence-level subtasks | | | | Frame-level subtasks | | | | | | | | |
|---|---|---|---|---|---|---|---|---|---|---|---|---|---|
| Model | Day (↓) | Time (↓) | Strain | Lights | Approach | Chase | Close | Contact | Huddle | O/E | O/G | O/O | Watching |
| PCA | 0.09416 | 0.09445 | 51.60 | 54.65 | 0.86 | 0.14 | 49.27 | 37.87 | 12.71 | 0.21 | 0.60 | 0.53 | 6.65 |
| TVAE | 0.09403 | 0.09442 | 52.98 | 56.80 | 1.07 | 0.45 | 59.33 | 44.77 | 21.96 | 0.27 | 0.83 | 0.62 | 10.20 |
| T-BYOL | 0.09362 | 0.09373 | 60.95 | 65.05 | 1.68 | 0.72 | 62.48 | 48.19 | 18.52 | 0.35 | 0.96 | 0.82 | 17.77 |
| T-BERT | 0.09262 | 0.09276 | 78.63 | 68.84 | 1.80 | 0.87 | 70.22 | 55.84 | 30.24 | 0.51 | 1.40 | 1.12 | 17.27 |
| TS2Vec | 0.09380 | 0.09422 | 57.12 | 65.60 | 1.29 | 0.66 | 59.53 | 46.13 | 24.74 | 0.35 | 1.09 | 0.74 | 12.37 |
| T-Perceiver | 0.09322 | 0.09323 | 69.81 | 69.68 | 1.57 | 1.27 | 60.84 | 47.81 | 28.32 | 0.41 | 1.16 | 0.86 | 16.42 |
| T-GPT | 0.09269 | 0.09384 | 64.45 | 65.39 | 1.73 | 0.64 | 69.05 | 55.78 | 23.80 | 0.46 | 1.12 | 1.05 | 17.86 |
| T-PointNet | 0.09275 | 0.09320 | 66.01 | 67.15 | 2.56 | **4.57** | **70.68** | **55.96** | 21.23 | **0.84** | **2.79** | **2.32** | 15.61 |
| BAMS | **0.09094** | **0.08989** | **88.23** | **72.00** | **2.74** | 1.89 | 67.22 | 53.43 | **31.43** | 0.59 | 1.61 | 1.57 | **18.15** |
| ↳ short-term | 0.09288 | 0.09294 | 61.33 | 66.34 | 1.80 | 1.15 | 66.58 | 52.60 | 25.34 | 0.39 | 1.09 | 0.98 | 16.74 |
| ↳ long-term | 0.09174 | 0.09037 | 86.49 | 70.91 | 2.10 | 1.06 | 61.99 | 49.12 | 29.09 | 0.45 | 1.32 | 1.08 | 13.98 |

Table 2: *Linear readouts of mouse behavior.* We report the BAMS against the top performing models in the MABe 2022 challenge. All numbers are reported in (59) except for TS2Vec and T-BYOL, which we produce. The scores show the performance of the linear readouts across 13 different tasks. Mean-squared error (MSE) is used in the case of tasks 1 and 2, since they are continuously labeled. In the rest of the subtasks, which are binary (yes/no), F1-scores are used. The best-performing models are those with low MSE scores and high F1-scores, are highlighted in bold.

and $\mathbf{z}_{t,3}$, for each mouse respectively and concatentate the embeddings for all three individuals to build a joint embedding for evaluation. The model is trained for 500 epochs using the Adam optimizer with a learning rate of $10^{-3}$. More details can be found in Appendix B.

**Interaction loss.** To learn additional features that are useful in predicting animal interactions for multi-agent settings, we introduce a simple auxiliary loss to predict the distances between the trio at time $t$. Our input features do not include any information about the global position of the mice in the arena, so the model can only rely on the inherent behavior and movement of each individual mouse to draw conclusions about their proximity. Thus, we build a network $h$ that takes in the embeddings of two mice $i$ and $j$ and predicts the distance $d_{i,j}$ between them as follows,

$$\mathcal{L}_{\text{aux}} = \|h(\mathbf{z}_{t,i}, \mathbf{z}_{t,j}) - \mathrm{d}_{i,j}\|_2^2. \tag{6}$$

This penalty is added to the loss in Equation 5 to encourage learning of shared features across the different individual embeddings.

**Evaluation protocol.** To evaluate the performance of the model in detecting frame- and sequence-level behaviors, we compute representations $\mathbf{z}_{t,1}$, $\mathbf{z}_{t,2}$ and $\mathbf{z}_{t,3}$ for each mouse respectively, which we then aggregate into a single mouse triplet embedding using two different pooling strategies. First, we apply average pooling to get $\mathbf{z}_{t,\text{avg}}$. Second, we apply max pooling and min pooling, then compute the difference to get $\mathbf{z}_{t,\text{minmax}} = \mathbf{z}_{t,\text{max}} - \mathbf{z}_{t,\text{min}}$. Both aggregated embeddings are concatenated into a 128-dim embedding for each frame in the sequence. Evaluation of each one of the 13 tasks, is performed by training a linear layer on top of the frozen representations, producing a final F1 score or a mean squared error depending on the task.

### 3.2.2 Results

We compare our model against PCA, trajectory VAE (TVAE) (13), TS2Vec (69), BYOL (17), and the top performing models in the MABe2022 challenge (61) which are adapted respectively from Perceiver (23), GPT (7), PointNet (49) and BERT (14). All methods are trained using some form of reconstruction-based objective, with both T-Perceiver and T-BERT using masked modeling. Both T-BERT and T-PointNet also supplement their training with a contrastive learning objective using positives from the same sequence. It is also important to note that the training set labels from two tasks (Lights and Chase) are made publicly available, and are used as additional supervision in the T-Perceiver and T-BERT models. We do not use any supervision for BAMS.

Our model achieves a new state-of-the-art result on the MABe Multi-Agent Behavior 2022 - Mouse Triplets Challenge, as can be seen in Table 2. We rank first overall based upon our performance on all tasks, and show impressive boosts in performance on global sequence-level tasks where we find a 22% improvement in the Strain task and 5% improvement in the Light task. In the frame-level tasks, we remain competitive with other approaches, and rank 1st on 3 out of 9 of the frame-level subtasks; this is in contrast to the other top performing model that explicitly models the interaction between mice by introducing hand-crafted pairwise features. This is outside the scope of this work, as we do not focus on social interactions beyond predicting the distance between mice.

We observe that our proposed method results in significant gaps on sequence-level tasks. In particular, we observe a marked improvement on the prediction of strain over the second-place model at 78.63%, with BAMS yielding a 10% improvement in accuracy at 88.23%. These big improvements suggest that we might have identified and addressed a critical problem in behavior modeling. In the next section, we conduct a series of ablations to further dissect our model's performance.

Multi-timescale embedding separation also enables us to probe our model for timescale-specific features. In Table 2, we report the decoding performance with short-term embeddings and long-term embeddings respectively. We find that sequence-level behavioral factors are better revealed in the long-term space, while the frame-level factors are more distinct in the short-term space. That being said, decoding performance is still best when using both timescales.

By default, BAMS is pre-trained with all available trajectory data. We also test BAMS in the inductive setting, where we only pre-train using the training split (only 1800 out of 5336) of the dataset. Results are reported in Appendix B.5. We find that the performance drop is modest, and that BAMS trained in this setting still beats all other methods.

### 3.2.3 Ablations

To understand the role of the different proposed components in enabling us to achieve state-of-the-art performance, we conduct a series of ablations on the MABe benchmark that we report in Table 3. First, we compare the performance of BAMS when trained with the traditional reconstruction-based objective (multi-step sequential prediction). BAMS without the `HoA` prediction objective, performs comparably with many of the top-entries, though performance is ultimately improved when using this novel learning objective. We note that we had to reduce the number of prediction frames to 10, as the training fails if we go beyond. With our `HoA` objective, we can use 30

| Hist. of actions | Bootstrapping | Multi-timescale | Seq MSE | Seq F1 | Frame F1 |
|:---:|:---:|:---:|:---:|:---:|:---:|
| ✓ | ✓ | ✓ | **0.090415** | **80.12** | **19.85** |
|   | ✓ | ✓ | 0.093100 | 68.30 | 19.46 |
| ✓ |   | ✓ | 0.090717 | 78.11 | 19.45 |
| ✓ | ✓ |   | 0.092483 | 73.37 | 19.52 |

Table 3: *Ablations on the Mouse Triplet Dataset.* We report the average sequence-level MSE and F1-score, and the average frame-level F1-score.

frames, which strongly suggests that this loss can be stably applied over longer time horizons when compared to other losses.

Next, we analyze the role of the multi-timescale bootstrapping in improving the quality of the representation. When removing the bootstrapping objective, we find a 2% drop in the sequence-level averaged F1-score, as well as modest drops in frame-level performance. This suggests that this learning objective may help to resolve global and intermediate-scale features. We also ablate the multi-timescale component, i.e. we perform bootstrapping but in the same space, by using a single TCN that spans the receptive field of the two used originally. This results in sub-optimal performance, which emphasizes the idea that having different objectives in different spaces is critical to prevent interference and provide ideal performance. We report additional ablations in Appendix B.4, including an ablation of the interaction module where we find that it is not critical for good performance.

## 3.3 Experiments on Fruit Fly Groups Dataset

### 3.3.1 Experimental Setup and Tasks

The fruit fly groups dataset is the second dataset in the MABe benchmark. It consists of tracking data of a group of 9 to 11 flies interacting in a small dish (Figure 5). Tracking data consists of 19 keypoints on each fly body and wings, and is recorded at a much higher frame rate compared to the mice (150 vs. 30 fps). Precise neural activity manipulations are performed on certain neurons which, when activated, induce certain types of behavior including courtship, avoidance and female aggression (61). Additionally, the groups of flies are differentiated by various genetic mutations and tagged by sex. This along with other behavioral factors provide us with 50 different subtasks, both frame-level and sequence-level, that can be use to evaluate the learned representations.

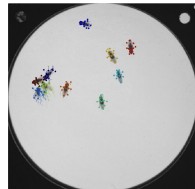

Figure 5: *Sample frame from the Fruit Fly Groups Dataset.*

In this set of experiments, we are interested in testing whether our proposed method generalizes to a dataset from another organism, and whether it can work out-of-the-box with minimal expert

intervention. In particular we chose to not extract any features manually and only use the provided data, we use the default hyperparameters found in the previous experiment and do not perform hyperparameter tuning, and finally we do not use an interaction module. Note that we follow the same evaluation procedure established previously.

### 3.3.2 Results

We compare our model against PCA, TVAE, TS2Vec, and the top performing models in the fly challenge: T-GPT, and T-Perceiver. Our model achieves state-of-the-art performance on the fly dataset, as seen in Table 4. BAMS outperforms other models on both frame-level and sequence-level sub-tasks, and we note a significant boost in the average frame-level F1 score. This result further demonstrates the generalizability of our approach to new datasets and scaling to an even larger numbers of animals.

| Model | All F1 | Seq F1 | Frame F1 |
|---|---|---|---|
| PCA | 42.5 | 23.0 | 45.2 |
| TVAE | 37.0 | 22.2 | 39.0 |
| T-Perceiver | 44.8 | 19.7 | 48.2 |
| T-GPT | 45.8 | 24.5 | 48.7 |
| BAMS | **48.2** | **25.4** | **51.1** |

Table 4: *Linear readouts of fly behavior.* We report the average performance of various models on both frame-level, and sequence-level subtasks.

## 4 Related Work

### 4.1 Animal behavior analysis

**Pose estimation and animal tracking.** Recently, there has been a recent democratization of automated methods for pose estimation and animal tracking that has made it possible to conduct large scale behavioral studies in many scientific domains. These tools abstract behavior trajectories from video recordings, and facilitate the modeling of behavioral dynamics. Most pipelines for analyzing animal behavior consist of three key steps (36): (1) pose estimation (60; 67; 15; 48; 40), (2) spatial-temporal feature extraction (35; 10), and (3) quantification and phenotyping of behavior (43; 39; 66). In our work, we consider the analysis of behavior after pose estimation is performed. However, one could imagine using our multi-timescale bootstrapping approach for representation learning in video analysis, where self-supervised losses have been proposed recently for keypoint discovery (60; 64; 24).

**Disentanglement of animal behavior in videos.** In recent work (54; 32; 63), two distinct disentangled behaviorial embeddings are learned from video, separating non-behavioral features (context, recording condition, etc.) from the dynamic behavioral factors (pose). This has even been applied to situations with multiple individuals, performing disentanglement on each individual (21). This is performed by training two encoders, typically variational autoencoders (VAEs) (29), on either multi-view dynamic information or a single image. These encoders create explicitly separated embeddings of behavioral and context features. In comparison to this work, rather than seperating behavior from context, our model considers the explicit separation of behavioral embeddings across multiple timescales, and considers the construction of a global embedding that is consistent over long timescales.

**Modeling social behavior.** For social and multi-animal datasets, there are a number of other challenges that arise. Simba (45) and MARS (52) have similar overall workflows for detecting keypoints and pose of many animals and classifying social behaviors. More recently, a semi-supervised approach TREBA has been introduced (59) for building behavior embeddings using task programming. TREBA is built on top of the trajectory VAE (13), a variational generative model for learning representations of physical trajectories in space. In our work, we do not consider a reconstruction objective but a future prediction objective, in addition to bootstrapping the behavior representations at different timescales.

### 4.2 Representation learning for sequential data

The self-supervised learning (SSL) framework has gained a lot of popularity recently due to its impressive performance in many domains (14; 9; 12). Many SSL methods are built based on the concept of *instance-specific* alignment loss: Different views of each datapoint are created based on pre-selected augmentations, and the views that are produced from the same datapoint are treated as positive examples; while the views that are produced from different datapoints are treated as negative examples. While contrastive methods like SimCLR (9) utilize both positive examples and negative examples to guide the learning, BYOL (17) proposes a framework in which augmentations of a

sample are brought closer together in the representation space through a predictive regression loss. Recent work (18; 50; 44; 2) applies BYOL to learn representations of sequential data. In such cases, neighboring samples in time are considered to be positive examples of each other, assuming temporal smoothness of the semantics underlying the sequence. The model is trained such that neighboring samples in time are mapped close to each other in the representation space. However, these methods use a single scale to define similarities unlike our method.

Recently, self-supervised methods such as TS2vec (69) learn self-supervised representations for sequential data by generating positive sub-sequence views through more complex temporal augmentations that can be integrated at instance or local level. Positive sub-sequences are temporally contrasted with other representations within the same sequence as well as contrasted with representations of other available sequences. However, unlike our method, the same space of representations are used for learning these multi-scale representations. We learn a separate space of representations for different time scales to encourage disentanglement.

The idea of using multi-scale feature extractors can be found in representation learning. In (50), a video representation learning framework, two different encoders process a narrow view and a broad view respectively. The narrow view corresponds to a video clip of a few seconds, while the broad view spans a larger timescale. The objective, however, is different to ours, as the narrow and broad representations are brought closer to each other, in the goal of encoding their mutual information. This strategy is also used in graph representation learning where a local-neighborhood of node is compared to its global neighborhood (19; 22). Our method differs in that we bootstrap the embeddings at different timescales separately, this is important to maintain the fine granularity specific to each timescale, thus revealing richer information about behavioral dynamics.

## 5   Conclusion

Behavior is likely to be driven by a number of factors that can unfold over different timescales. Thus, having ways to model behavior and discover differences in behavioral repertoires or actions at many scales could provide insights into individual differences, and help, for example, detect signatures of cognitive impairment (66). We make steps towards addressing these needs by proposing BAMS, a novel self-supervised approach that learns representations for behavioral data at different timescales.

Our analysis on realworld datasets centers on two different datasets in a multi-agent benchmark, both with very different dimensionalities and variable tasks. Despite this, our approach is designed to model a wide range of behaviors, whether it's individual, social, structured or naturalistic. More multi-scale and multi-task benchmarks like MABe are needed in order properly evaluate and guide the development of tools for general behavior analysis.

Currently, our model has a relatively simple way of (optionally) modeling interactions between animals in the multi-agent setting. Despite this, we show really competitive performance in multi-agent analysis without using handcrafted interaction features like T-PointNet or T-BERT (61). Moving forward, it would be interesting to develop new ways to learn features of multi-animal interactions, especially in open environments where animals might come in and out of frame or get occluded.

Our experiments highlight the truly multi-scale nature of BAMS and show that our method can learn to distinguish global as well as temporally local behaviors. Currently, we seperate short-term from long-term dynamics through a contrastive loss and separation of the information into two latent spaces for each scale. Although this approach appears to be effective, we don't modify the loss explicitly to disentangle the two latent spaces. By providing additional incentives for the model to separate short from long-term dynamics we hope to improve the interpretability of the model.

When extending our model from mice to flies, we found that it was possible to use the same overall model architecture and hyperparameters, despite the major differences in datasets and underlying tasks. Thus, given the robustness of the method, we imagine that it can be utilized in the analysis of other species and even more diverse types of behavior (42; 26). By combining our modeling approach with methods for self-supervised video keypoint discovery (57), we could further extend BAMS to raw video data without needing the intermediate step of pose estimation. We also look forward to the opportunities provided by simultaneous recordings of the brain and behavior (30; 47) and other multi-modal sources of input that could be leveraged to further study how behavior unfolds across different timescales.

## Acknowledgements

We would like to thank Mohammad Gheshlaghi Azar and Remi Munos for their feedback on the work. This project was supported by NIH award 1R01EB029852, NSF awards IIS-2039741 and IIS-2146072, as well as generous gifts from the Alfred Sloan Foundation, the McKnight Foundation, and the CIFAR Azrieli Global Scholars Program.

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

# A Experimental details: Simulated Quadrupeds

## A.1 Data generation

**Simulation details.**    We record a total of 5182 trajectories. 2756 were generated for robots of type ANYmal B and 2426 sequences were generated for robots of type ANYmal C. These are quadruped robots, which means that they have four legs. Each leg has 3 degrees of freedoms - hip, shank and thigh. The position and velocities of these degrees of freedom for all 4 legs were recorded. This results in 24 features for each robot. Robots are generated while traversing an procedurally generated environment with different terrain types and traversal difficulty, as show in Figure 6. We only keep trajectories that correspond to a successful traversal.

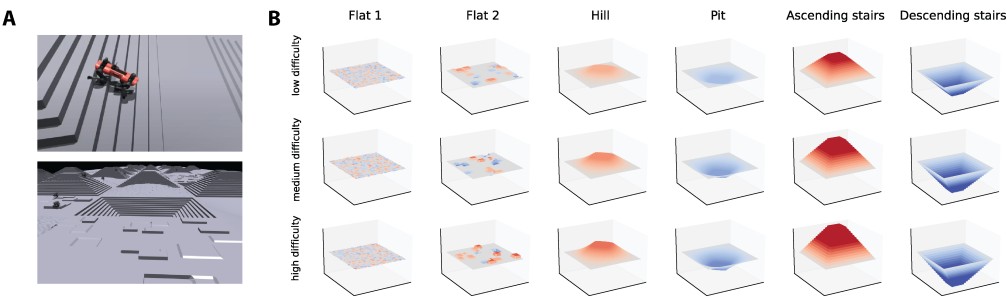

Figure 6: *Visualization of different simulated environments*. (A) Screenshots from the simulator showing a robot walking down some stairs, and a view of the terrain landscape. (B) Visualization of the different terrain sections, characterized by a terrain type and different levels of difficulty. Terrain are made more difficult to traverse by either making them more rough or have steeper slopes.

**Tasks.**    To evaluate the representation quality of our model, we use multi-task probes that correspond to different long-term and short-term behavioral factors.

- Robot type: the robot can either be of type "ANYmal B" or "ANYmal C". These robots have the same degrees of freedom and tracked joints but differ by their morphology. This is a sequence-level task.
- Linear velocity: the command of the robot is a constant velocity vector. The amplitude of the velocity dictates how fast the robot is commanded to traverse the environment. A higher velocity would translate into more clumpsy and more risk-taking behavior. This is a sequence-level task.
- Terrain type: the environment is generated with multiple segments of five terrain types that are categorized as: flat surfaces, pits, hills, ascending and descending stairs. This is a frame-level task.
- Terrain slope: the slope of the surface the robot is walking on. This is a frame-level task.
- Terrain difficulty: the different terrain segments have different difficulty levels based on terrain roughness or steepness of the surface. This is a frame-level task.

**Why this dataset.**    Simulation-based data collection enables access to information that is generally inaccessible or hard to acquire in a real-world setting. Unlike noisy measurements coming from the camera-based feature extractor in the case of the mouse dataset, physics engines do not suffer from the problem of noise. Instead, they provide accurate ground-truth information about the creature and the world state free of charge. Access to such information is at times critical for scrutinizing the capabilities of the learning algorithms.

## A.2 Visualizing differences between short-term and long-term embeddings

In Figure 7, we visualize how the short-term and long-term embeddings evolve over time, for a single sample sequence. We note a clear difference in the smoothness in the two timescales. In the short-term embeddings, we note a clear block structure corresponding to different blocks of behavior that span a few seconds, while in the long-term embeddings the representation is more stable over

time. This suggests that, as expected, the bootstrapping objectives are forming representations with different levels of granularity.

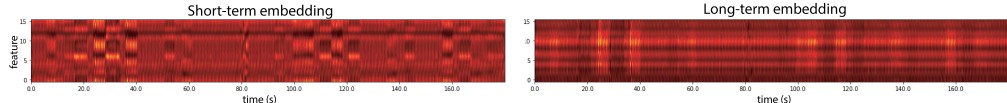

Figure 7: *Visualization of the short-term and long-term embeddings.* We visualize for a single sequence how the short-term and long-term embeddings evolve over time.

# B    Experimental details: Mouse Triplet

## B.1    Feature extraction

Each mouse in the arena is tracked using 12 anatomically defined keypoints. We process these keypoints to extract 36 different features characterizing each mouse individually, similar to (52). We separate the keypoints into two different areas, the head and the body, for each we extract different measures of displacement, that we express in the frame of the mouse, i.e. these features are invariant to the pose of the mouse relative to the arena. These features include:

- Head linear velocity vector that we express using polar coordinates.
- Head angular velocity denoting the change in the heading direction in the arena.
- Body linear velocity vector that we express using polar coordinates.
- Body angular velocity denoting the change in the direction of the body with respect to the arena.
- Angular and linear velocities of the fore paws and the hind paws.
- Spine length change, depicting the expansion and contraction of the mouse's body.
- Angles formed by the tail with respect to the body.

We normalize all features before training. We also use cosine and sinus of the angles instead of the angles. During training, we did not use any form of augmentation.

**Noise in the data.**    Because of errors in pose estimation and tracking, there are sometimes errors in the tracking data, notably some identity swap issues (61). To address this, we simply zero out all of the corresponding features and flag the frame as invalid. A binary feature is also add to the input features indicating whether or not the frame is valid. When predicting future actions, we only compute the error over windows in which at least 80% of the frames are valid.

## B.2    Difference between Histogram of Actions and previous objectives

Our novel objective consists in predicting the future histogram of actions instead of predicting the future sequence of actions. In Figure 8, we show what the target is for a sample from the MABe Mouse Triplet dataset. Note that the time dimension is collapsed, blurring the exact unrolling of the future events, but preserving the set of values that these actions will sweep. Note that the loss (EMD) is applied for each action feature.

We show that by relaxing our future prediction loss to a HoA loss, we benefit in terms of the representations that are learned by the model, especially for sequence-level (global tasks). Thus, in many ways, we show that directly forecasting, which is what many previous approaches have used for representation learning, can actually lead to representations that capture less of the task structure. While the model doesn't predict future timesteps directly, we can visualize the histogram prediction for the model and ground truth (Figure 9).

## B.3    Training details

**Architecture.**    We use two TCNs (4). Each TCN is built using multiple residual blocks, each residual block is composed of two convolutional layers, and use PReLU activation, dropout and weight normalization. All convolutions are dilated with a rate $r$, that increases after each residual

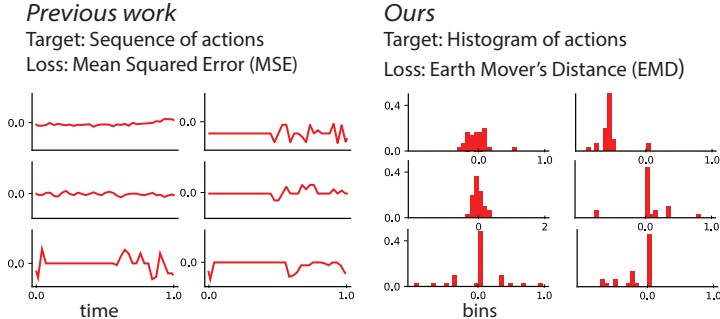

Figure 8: Prediction target for a sample of the MABe Mouse Triplet dataset.

block. The formula is $r^i$ where $i$ is the index of the residual block. The first TCN is the short-term encoder, which uses 4 blocks with output sizes $[64, 64, 32, 32]$ and a dilation rate $r = 2$. The second TCN is the long-term encoder, which uses 5 blocks with output size $[64, 64, 64, 32, 32]$ and a dilation rate $r = 4$. The output of both encoders are concatenated to form a $64d$ embedding. The predictor is a multi-layer perceptron (MLP) that has 4 hidden layers.

**Training.** We train the model for 500 epochs using the Adam optimizer with a learning rate of $10^{-3}$ and weight decay $4 \cdot 10^{-5}$, we decrease the learning rate to $10^{-4}$ after 100 epochs. We use a batch size of 96, and compute the future histogram of action prediction error for each timestep $t$ starting at 5 seconds after the start of each sequence, in order to allow the model to aggregate enough context. We set the learning rate of the predictors used for bootstrapping to be 10 times higher than the learning rate used for the rest of the weights.

**Evaluation.** During the development of the model (Figure 10), we test our model on the public test splits, and only look at the performance on the private set after finishing any hyperparameter tuning. We repeat the training and evaluation 5 times and report the average performance

### B.4 Additional ablations

In addition to those mentioned in the main text, we perform two additional ablations to BAMS (Table 5). In our first experiment, we removed the interaction loss from the model, meaning that the

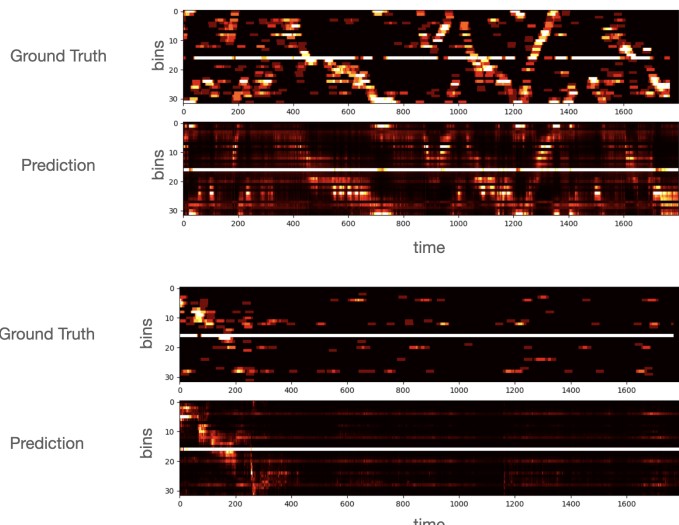

Figure 9: Visualization of histograms of future actions, for two random action features. For a timestamp t on the time axis, we show the 32 dimensional histogram of future actions, which is the target of prediction for BAMS.

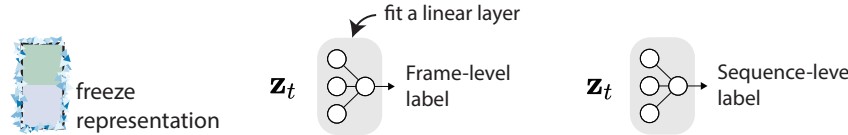

Figure 10: Linear evaluation protocol. The model is frozen, and for each task, a single linear layer is trained to predict the corresponding labels.

dynamics of each mouse are modeled completely independently from each other. The ablated model sees a small drop in performance, but continues to outperform all other methods on average.

## B.5 BAMS in the inductive setting

The mouse triplet dataset (5336 sequences) has three different sets, a training set (1800 sequences), a private test set and a public test set. During training of the representation learning model, we can either pre-train on all of the available data (transductive setting) or on the training set only (inductive setting). During linear evaluation, the different linear layers are trained using labels from the training set and the performance is reported on the public test set (during the challenge) and then on the private test set (to rank models).

We train BAMS in the inductive setting and report the performance in Table 5. We find that even when BAMS is trained with approximately one third of the data, the drop in performance is modest. More importantly, BAMS preserves its ranking compared to other methods, and still achieves state-of-the-art performance.

Table 5: *Linear readouts of mouse behavior.* The best-performing models are those with low MSE scores and high F1-scores.

| | Sequence-level subtasks | | | | Frame-level subtasks | | | | | | | | |
|---|---|---|---|---|---|---|---|---|---|---|---|---|---|
| Model | Day ($\downarrow$) | Time ($\downarrow$) | Strain | Lights | Approach | Chase | Close | Contact | Huddle | O/E | O/G | O/O | Watching |
| PCA | 0.09416 | 0.09445 | 51.60 | 54.65 | 0.86 | 0.14 | 49.27 | 37.87 | 12.71 | 0.21 | 0.60 | 0.53 | 6.65 |
| TVAE | 0.09403 | 0.09442 | 52.98 | 56.80 | 1.07 | 0.45 | 59.33 | 44.77 | 21.96 | 0.27 | 0.83 | 0.62 | 10.20 |
| T-BERT | 0.09262 | 0.09276 | 78.63 | 68.84 | 1.80 | 0.87 | 70.22 | 55.84 | 30.24 | 0.51 | 1.40 | 1.12 | 17.27 |
| TS2Vec | 0.09380 | 0.09422 | 57.12 | 65.60 | 1.29 | 0.66 | 59.53 | 46.13 | 24.74 | 0.35 | 1.09 | 0.74 | 12.37 |
| T-Perceiver | 0.09322 | 0.09323 | 69.81 | 69.68 | 1.57 | 1.27 | 60.84 | 47.81 | 28.32 | 0.41 | 1.16 | 0.86 | 16.42 |
| T-GPT | 0.09269 | 0.09384 | 64.45 | 65.39 | 1.73 | 0.64 | 69.05 | 55.78 | 23.80 | 0.46 | 1.12 | 1.05 | 17.86 |
| T-PointNet | 0.09275 | 0.09320 | 66.01 | 67.15 | 2.56 | 4.57 | 70.68 | 55.96 | 21.23 | 0.84 | 2.79 | 2.32 | 15.61 |
| BAMS - no interaction | 0.09164 | 0.09154 | 83.47 | 71.23 | 2.55 | 2.03 | 63.63 | 50.97 | 31.15 | 0.58 | 1.47 | 1.37 | 15.10 |
| BAMS - inductive | 0.09112 | 0.09132 | 83.44 | 70.39 | 2.62 | 1.40 | 65.98 | 52.39 | 31.08 | 0.60 | 1.54 | 1.40 | 18.14 |
| BAMS - transductive | 0.09094 | 0.08989 | 88.23 | 72.00 | 2.74 | 1.89 | 67.22 | 53.43 | 31.43 | 0.59 | 1.61 | 1.57 | 18.15 |

Table 6: *TS2Vec Linear readouts of mouse behavior.*

| | Sequence-level subtasks | | | | Frame-level subtasks | | | | | | | | |
|---|---|---|---|---|---|---|---|---|---|---|---|---|---|
| Model | Day ($\downarrow$) | Time ($\downarrow$) | Strain | Lights | Approach | Chase | Close | Contact | Huddle | O/E | O/G | O/O | Watching |
| TS2Vec-I | 0.09380 | 0.09422 | 57.12 | 65.60 | 1.29 | 0.66 | 59.53 | 46.13 | 24.74 | 0.35 | 1.09 | 0.74 | 12.37 |
| TS2Vec-T | 0.09882 | 1.0252 | 45.82 | 46.69 | 0.72 | 0.14 | 45.19 | 34.93 | 9.38 | 0.186 | 0.38 | 0.38 | 05.31 |
| TS2Vec-IT | 0.09846 | 1.01646 | 46.67 | 44.28 | 0.67 | 0.13 | 44.56 | 33.87 | 9.79 | 0.178 | 0.42 | 0.42 | 04.58 |

## B.6 Notes on TS2Vec experiments

TS2Vec (69) employs two types of contrastive losses to learn representations. The first of these losses is an instance contrastive loss which contrasts a sequence with all other sequences in a batch which are treated as negative examples, while two subsequences extracted from the same sequence are treated as positive examples. The second loss is a temporal contrastive loss which acts along a single time series. Temporal representations of nearby time points are taken as positive examples, while the rest of the time points within the same sequence are taken as negative examples. The results for the three versions of TS2vec, namely TS2Vec-I, which uses only instance contrastive loss, TS2Vec-T, which uses only temporal contrastive loss,and TS2Vec IT, which uses both instance and temporal contrastive losses, are listed in Table 6. Our TS2Vec experiments on the mouse dataset showed that using temporal contrastive loss resulted in worse performance across all tasks as compared to only

using instance contrastive loss. For this reason, we only report results for TS2vec that only employs instance contrastive loss.

We note that for both TS2Vec and TS2Vec-IT, we ran into out-of-memory errors when creating instance-level or global contrast. Contrastive learning methods usually incur high computational costs, we find that our method, which doesn't rely on negative examples, can scale better when working with longer sequences and larger datasets.

