# OpenReview forum: "Relax, it doesn’t matter how you get there: A new self-supervised approach for multi-timescale behavior analysis"
_NeurIPS.cc/2023/Conference — NeurIPS 2023 spotlight_

### Official Review · Reviewer_7X95 · 2023-07-05

**Soundness:** 2 fair
**Presentation:** 3 good
**Contribution:** 2 fair
**Rating:** 5
**Confidence:** 3

**Summary:**

The authors propose BAMS, an architecture for learning agent behaviors by mapping observations to actions. BAMS utilizes two Temporal Convolutional Networks (TCNs) operating at different time scales, enabling modeling of both short-term and long-term trends in data through varying dilation rates. The short-term and long-term representations obtained from these networks are further enhanced using self-supervised bootstrapping objectives inspired by previous research. In addition, the authors introduce a novel decoder that predicts histograms to capture distributions instead of discrete predictions during training, which they believe aids in long-term forecasting. For deployment, a linear layer is fitted on the learned representations, followed by local training on the linear layer prior to evaluation. Experimental results on mobile robotics and animal behavior datasets demonstrate promising predictions compared to baseline methods.

**Strengths:**

These are the strengths of the paper in my opinion:

Clear and Coherent Writing: The paper is well-written and maintains a clear and organized structure, ensuring a pleasant reading experience.

Relevance to the Research Community: The problem of learning representations for multi-time scale behaviour addressed in the paper is very relevant to the research community.

Novel Approach Of predicting Histograms: The paper introduces an interesting innovative approach by utilizing histograms for predictions to capture distributions which are accompanied by the introduction of compatible loss functions. I think this idea would help with capturing multi-modal behaviours/predictions in a better manner for future applications, though further analysis needs to be done in this regard.

Strong Empirical Performance: The proposed method demonstrates strong empirical performance on two challenging benchmarks. By achieving promising results in these real-world scenarios, the paper highlights the effectiveness and potential practical application of the approach.

**Weaknesses:**

The weakness of the paper under review is listed below:

Lack of Well-Founded Design Principles: The design choices in the proposed method appear to be based on arbitrary experimentation rather than being guided by well-established principles. This absence of a solid theoretical foundation raises concerns about the method's reliability and may compromise its technical rigour/novelty.

Limited Applicability to General Cases: I doubt the method's suitability for the general case of learning and predicting long-term behaviours in time series or sequential data, which is more relevant to the machine learning community. The reliance on specific datasets and the lack of comprehensive exploration across diverse scenarios raises questions about the method's ability to generalize its performance beyond the limited contexts it has been tested on. More empirical evidence is needed to establish its applicability in broader settings.

Insufficient Discussion of Related Works: The paper fails to adequately address and discuss related works on hierarchical neural network architectures [2][3] that are specifically designed to model multi-time scale behaviour. This omission prevents a comprehensive understanding of the method's positioning and effectiveness compared to existing "generic" approaches. It would be valuable to include discussions and comparisons with these alternative methods to provide a more complete perspective.

Cumbersome Use of Multiple Loss Functions and Architecture Modification: The method's reliance on multiple loss functions and the need to modify the architecture during deployment by introducing and training a linear decoder may be perceived as cumbersome and lacking elegance. This complexity raises concerns about the practicality and efficiency of the proposed approach, as simpler and more streamlined alternatives exist. A more streamlined and efficient design would enhance the method's overall appeal.

**References**

[2] Chung, Junyoung, Sungjin Ahn, and Yoshua Bengio. "Hierarchical multiscale recurrent neural networks." arXiv preprint arXiv:1609.01704 (2016).

[3] Mujika, Asier, Florian Meier, and Angelika Steger. "Fast-slow recurrent neural networks." Advances in Neural Information Processing Systems 30 (2017).

**Questions:**

Applicability to General Time Series/Dynamics Modelling: Do you believe that your method is applicable to a wide range of time series/dynamics modelling tasks, thus increasing its relevance to the broader machine learning community? It would be valuable to understand the potential generalizability and versatility of your method beyond the specific datasets and scenarios discussed in the paper.

Differentiation from Existing Multi-Time Scale Modelling Literature [2][3]: How does your method stand out when compared to the existing literature on multi-time scale modelling? It would be insightful to highlight the unique contributions or advantages that set your approach apart from other techniques designed for similar purposes.

Long-Horizon Predictions and Forecasting: How accurately can BAMS make long-horizon or multistep-ahead predictions/forecasting, especially for the robotic dataset? This information was not very clear / lacking in the paper. If long-horizon predictions are feasible, what is the reliable prediction horizon (e.g., number of seconds into the future) that your method achieves on both datasets without performance degradation? Did you make a plot of the predictions vs ground truths for both datasets, what did this plot convey ?? This information would provide valuable insights into the method's capability for long-term forecasting.

**References**

[2] Chung, Junyoung, Sungjin Ahn, and Yoshua Bengio. "Hierarchical multiscale recurrent neural networks." arXiv preprint arXiv:1609.01704 (2016).

[3] Mujika, Asier, Florian Meier, and Angelika Steger. "Fast-slow recurrent neural networks." Advances in Neural Information Processing Systems 30 (2017).

**Limitations:**

The authors were upfront about some limitations of the work. There was no discussion on broader impacts.

---

> ### Author Rebuttal · Authors · 2023-08-10
>
> Thank you for your comments and questions! We provide point-by-point responses below.
>
> > 1. “The design choices in the proposed method appear to be based on arbitrary experimentation rather than being guided by well-established principles. This absence of a solid theoretical foundation raises concerns about the method's reliability and may compromise its technical rigour/novelty.”
>
> **Reply:** We believe that our design choices were built from insights about the unique challenges of animal behavior analysis. We have extensively experimented with different representation learning objectives, and made critical observations about what can be improved in the current landscape of behavior representation learning methods. In particular we identify limitations of time series future prediction, as well as, the importance of multi-timescales modeling in multi-task settings. To tackle both of these challenges, we introduce a histogram prediction task for “relaxing” multi-step prediction, and (2) a multi-scale disentanglement loss to help separate the latents. We study the contribution of these different components through various ablations (Table 3), and find that both objectives are critical for state-of-the-art performance.
>
> > 2. “The method's reliance on multiple loss functions and the need to modify the architecture during deployment by introducing and training a linear decoder may be perceived as cumbersome and lacking elegance. [...] A more streamlined and efficient design would enhance the method's overall appeal.
>
> **Reply:** Throughout the paper, we utilize standard procedures commonly used in self-supervised learning [1,2], where we train the model without any labels and then use a linear decoder to evaluate the separability of the feature space from only a few labeled instances. Indeed, this is the evaluation strategy that is used in MABe, and all of the models are trained without labels and then a post-eval strategy is used to rank the methods in the competition and in the benchmark.
>
> Please note that Reviewer hofo notes that our post-hoc evaluation is a strength, and that BAMS “learns to characterize behavioral patterns, solely by post-training a linear classifier given the emergent latent codes” (hofo).” With this clarification over our evaluation procedure, we hope that you consider raising your score.
>
> > 3. “It would be valuable to understand the potential generalizability and versatility of your method beyond the specific datasets and scenarios discussed in the paper.”
>
> **Reply:** Analysis of behavior is becoming an increasingly important task in neuroscience, as well as advertising, marketing, and robotics. While we have not yet studied the application of BAMS in other time-series domains, we believe that many aspects of our approach could be more applied to a wide range of different datasets. In particular, the idea of relaxing the MSE or next-step prediction task to a HoA task could also be applied in other architectures and in different data domains. We think this is an exciting direction for future work.
>
> > 4. “Long-Horizon Predictions and Forecasting: How accurately can BAMS make long-horizon or multistep-ahead predictions/forecasting, especially for the robotic dataset? This information was not very clear / lacking in the paper. [...] Did you make a plot of the predictions vs ground truths for both datasets?.”
>
> **Reply:** While time series forecasting is an important task in general, in this work we are challenging the use of such an approach in cases where the downstream task is not forecasting. The focus of the paper is on representation learning and thus we use SSL and representation learning benchmarks and datasets rather than forecasting. We show that by relaxing our future prediction loss to a HoA loss, we benefit in terms of the representations that are learned by the model, especially for sequence-level (global tasks). Thus, in many ways, we show that directly forecasting, which is what many previous approaches have used for representation learning, can actually lead to representations that capture less of the task structure.
> While the model doesn’t predict future timesteps directly, we can visualize the histogram prediction for the model and ground truth (see the PDF uploaded to OpenReview).
>
> > 5. “The paper fails to adequately address and discuss related works on hierarchical neural network architectures [2][3] that are specifically designed to model multi-time scale behaviour”.
>
> **Reply:** In our discussion of related work, we focused more on the recent work in behavior modeling, and in temporal contrastive learning and bootstrapping which has been explored in recent years like for video [3] as an example, as we believed these papers to be most relevant to our current work and innovations.  In our revised paper, we plan to add more discussion of other architectures for multiscale sequence modeling and will include the references provided by the reviewer. Thank you for your suggestions!
>
> ---
>
> [1] Chen, Ting, et al. "A simple framework for contrastive learning of visual representations." ICML, 2020.
>
> [2] Sun, Jennifer J., et al. "MABe22: A Multi-Species Multi-Task Benchmark for Learned Representations of Behavior." ICML 2023.
>
> [3] Recasens, Adria, et al. "Broaden your views for self-supervised video learning." CVPR, 2021

---

> > ### Comment · Reviewer_7X95 · 2023-08-20
> >
> > Sorry for the delay in response.
> >
> > Thank you for your reply. I am not from a Neuroscience background but rather from a system identification/ dynamics learning/ deep sequential latent variable models background. To me, the intuition behind the ideas the authors propose like multi-time scale modelling and predicting a histogram over actions (a statistical measure over a patch of action sequence rather than directly reconstructing the actions) are interesting and justifiable as stated in the strengths section.
> >
> > But I think the methods (modelling assumptions/learning scheme/inference procedure) are rather ad-hoc and are fine-tuned to the specific application/competition, though it's commendable that these combinations result in state-of-the-art results in the particular use case.
> >
> > For example, I think the HoA procedure may be limiting to someone who wants to use the said method for a more general use case like time-series forecasting or learning forward dynamics models of any dynamical system (for control) where long-term predictions are important. More general-purpose models like transformers [1, 2,3] or recent advances in state space models [4, 5] which gave state-of-the-art performance in capturing long-range dependency on a variety of benchmarks are much broader in their potential application domains.
> >
> > Hence I retain my score of borderline accept. But I reduce my confidence score from 4 to a 3 due to my lack of background in Neuroscience and I can't judge its potential impact on the Neuroscience/Behaviour Modelling community.
> >
> > ------
> >
> > [1] Tom Brown, Benjamin Mann, Nick Ryder, Melanie Subbiah, Jared D Kaplan, Prafulla Dhariwal, Arvind Neelakantan, Pranav Shyam, Girish Sastry, Amanda Askell, et al. Language models are few-shot learners. Advances in neural information processing systems
> >
> > [2] Zhou, Haoyi, et al. "Informer: Beyond efficient transformer for long sequence time-series forecasting." AAAI 2021.
> >
> > [3] Nie, Yuqi, et al. "A time series is worth 64 words: Long-term forecasting with transformers." ICLR 2023.
> >
> > [4] Gu, Albert, Karan Goel, and Christopher Ré. "Efficiently modelling long sequences with structured state spaces." ICLR 2022
> >
> > [5] Smith, Jimmy TH, Andrew Warrington, and Scott W. Linderman. "Simplified state space layers for sequence modelling." ICLR 2023

---

### Official Review · Reviewer_Uc6V · 2023-07-06

**Soundness:** 3 good
**Presentation:** 3 good
**Contribution:** 2 fair
**Rating:** 6
**Confidence:** 5

**Summary:**

The authors propose a model called BAMS for unsupervised behavioral analysis. It combines two temporal convolutional neural networks with different timescales used to learn latents, as well as a loss to predicting a distribution of future actions. BAMS reaches strong performance on a novel, small-scale synthetic dataset as well as one of the MABE datasets.

**Strengths:**

1)	Paper well written and easy to follow.
2)	Simple combination of known features (TCN, distribution matching, BYOL) to reach strong performance for unsupervised behavioral analysis on two datasets.
3)	Ablation analysis is good, and gives insights into design choices.

**Weaknesses:**

1)	While the overall pipeline is novel, each component is rather well known and thus the overall novelty is somewhat limited.
2)	BAMS is only validated on one of two MABE datasets (the benchmark also contains flies), as well as one novel small synthetic dataset. While the results on the mouse MABE dataset are strong, how the method is adapted to multiple mice is rather ad-hoc. Thus, ideally it should be tested on single-animal behavioral datasets.
3)	For MABE the baselines are rather strong (as they are from the MABE competition), but they are mostly self-supervised methods using transformers (GPT, BERT, etc. style). It’s nice that the BAMS with TCNs can actually beat those methods for some of the behaviors – However, I think it would be great if a vanilla BYOL method could also be run by the authors to assess the novelty of BAMS.

**Questions:**

I wonder if the authors also ran BAMS on the flies in MABE?

**Limitations:**

In the abstract the authors state: “While some success has been found in building representations of animal behavior under constrained or simplified task-based conditions, many of these models cannot be applied to free and naturalistic settings where behavior becomes increasingly hard to model.”

I think this sentence should be removed, as this paper also does not deal with naturalistic behavior.

---

> ### Author Rebuttal · Authors · 2023-08-10
>
> Thank you for your comments and questions! We provide point-by-point responses below.
>
> > 1. “While the overall pipeline is novel, each component is rather well known and thus the overall novelty is somewhat limited.”
>
> **Reply:** The method consists of two core components: (1) a HoA predictive loss, (2) an architecture and contrastive loss to encourage multi-timescale disentanglement. While it is true that bootstrapping and temporal contrastive learning have been applied to sequential data (TS2Vec for example), HoA is a novel objective and we have not seen this idea applied to time series modeling. Through a series of ablations, we show how each component contributes to the overall accuracy, and these experiments point to the significance of our novel HoA loss formulation in providing critical boosts in performance.
>
> > 2. “While the results on the mouse MABE dataset are strong, how the method is adapted to multiple mice is rather ad-hoc. Thus, ideally it should be tested on single-animal behavioral datasets.”
>
> **Reply:**
> - Rationale behind selecting MABe:  One key innovation and motivation behind BAMS is its ability to build representations of behavior that allow for short-term (frame-level) classification as well as more global (sequence-level) classification. The Mabe dataset and benchmark provided a perfect fit for evaluation of diverse behaviors and behavioral traits that span different timescales. It is unique in that it contains both frame-level and sequence-level labels for behavior, and has sequences that are long enough to support separation of factors across timescales. For this reason, we found MABe to be a good fit for our analysis and few other datasets to meet the multitask criteria.
> - Ablations of the multi-mouse features: Because other methods we compare against use multi-mouse features, we use a simple interaction loss to capture some of the triplet-level behavior. To address your concern, we completely remove this component, and model each mouse independently: BAMS is trained on single animals in this case.   During evaluation we still need to use pooling to compare against other methods. We report the results in Table 1 in the accompanying pdf. We show that the model still ranks first despite not modeling multi-animal interactions.  We appreciate your comments and feel that with this ablation, the paper is stronger and the results are even more compelling.
>
>
> > 3. “For MABE the baselines are rather strong (as they are from the MABE competition), but they are mostly self-supervised methods using transformers (GPT, BERT, etc. style). [...] I think it would be great if a vanilla BYOL method could also be run by the authors to assess the novelty of BAMS.”
>
> **Reply:** Thanks for your suggestion. In addition to transformer-based architectures that are currently well ranked on the leaderboard, we also included a more recent time-series contrastive learning model, TS2Vec, as a baseline for Mabe, which uses both frame-level and sequence-level contrast. We thought that this was a strong contrastive baseline given that it was designed for time series data and also has an inherent mechanism for building local and global contrast. To address your concern, we ran a temporal version of BYOL [1] using a TCN backbone and  a single positivity window of around a minute.
>
> > 4. Question: I wonder if the authors tested the method on flies?
>
> **Reply:**   We haven’t yet tested on the flies dataset but think this would be an interesting dataset for future investigations.
>
> ---
>
> [1] Grill, Jean-Bastien, et al. "Bootstrap your own latent-a new approach to self-supervised learning." NeurIPS 2023

---

> > ### Comment · Reviewer_Uc6V · 2023-08-11
> >
> > Thanks for the rebuttal. I am happy that the other reviewers also appreciate the paper. Overall, due to a amount of benchmarking datasets:
> > 1) Just half of the MABE challenge was used (not the flies)
> > 2) While I agree that MABE is well suited for BAMS, due to the annotation at two different timescales; the generalization to multiple animals is ad-hoc in BAMS and a single animal benchmark should (also) have been tackled. Fundamentally, the behaviors in MABE are simple behaviors that can be scored by rule-based algorithms
> > 3) Despite my suggestion, you did not add another dataset in the rebuttal
> >
> > I did raise all those points in my original review. Thus, I will keep the score of 5, which, I believe, rather well summarizes this manuscript (let me paraphrase): technically solid paper with limited evaluation.

---

> > > ### Author Response · Authors · 2023-08-19
> > >
> > > Thank you for your response and engaging in the discussion period with us! We really appreciate your feedback.
> > >
> > > Per your suggestion, we tested BAMS on the Flies dataset in the Mabe Challenge. We are happy to report that we achieved state-of-the-art performance and are #1 on the leaderboard for this challenge (see results below)! We note that this result was obtained without any parameter tuning (using the default parameters in Mouse Triplets), and without using the interaction loss (i.e. BAMS is pre-trained to model individual-level behavior dynamics).
> > >
> > > |  | All F1 | Frame F1 | Sequence F1 |
> > > | - | - | - | - |
> > > | PCA | 42.5 | 23.0 | 45.2 |
> > > | TVAE | 37.0 | 22.2 | 39.0 |
> > > | T-Perceiver | 44.8 | 19.7 | 48.2 |
> > > | T-GPT | 45.8 | 24.5 | 48.7 |
> > > | BAMS (Ours) | **48.2** | **26.4** | **51.1** |
> > >
> > > As you can see, BAMS works well on both frame- and sequence-level tasks and outperforms the other baselines. We believe that the paper will be much stronger now with the inclusion of the flies dataset and the other experiments you suggested in the first round of responses.

---

> > > > ### Comment · Reviewer_Uc6V · 2023-08-21
> > > >
> > > > Congrats on the new MABE fly results. That's a great addition to the paper.
> > > >
> > > > I still think that more datasets (esp. individual animal datasets) should have been used and that the technical contribution is limited. However, I think it's a good paper that also clearly improved in the rebuttal phase. Overall, all reviewers had initially agreed that it should be an accept. With the hard work for the fly results, I would increase my score from 5 -> 6.

---

### Official Review · Reviewer_hofo · 2023-07-10

**Soundness:** 3 good
**Presentation:** 3 good
**Contribution:** 4 excellent
**Rating:** 7
**Confidence:** 4

**Summary:**

The paper introduces a self-supervised (autoregressive) learning system that processes behavioral data on two fixed time scales via TCNs. Combined, the TCN output it then used to predict behavioral histograms over a fixed time window. The innovative histogram encoding encourages the learning of behavioral distribution rather than exact behavioral sequence patterns, thus making the system independent of exact temporal dynamics. Additionally, a bootstrapping technique is used to encourage prediction locality (slightly unclear how exactly, see below). The results are quite impressive seeing that the system is not trained to actually classify or distinguish behavioral patterns or agent properties. The system indeed reaches state-of-the-art performance outperforming all other approaches in a recent mouse behavior analysis benchmark. Another artificial robot behavioral benchmark also yields great performance. In both cases the authors show that the emergent latent hidden state representations z learn to encode short- and long-term behavioral characteristics.

**Strengths:**

An autoregressive approach that learns to characterize behavioral patterns, solely by post-training a linear classifier given the emergent latent codes.

System outperforms state-of-the-art systems on a recent benchmark without supervised training.

Latent structures emerge that are very well-suited to characterize behavioral patterns (short-term dynamics) as well as more global behavioral tendencies (agent properties / time of day). They are – at least to a certain extent – directly interpretable.

An innovative histogram-based loss component that does most of the trick and should be applicable in other tasks and domains. (I wonder if you also attempted to use the more common KL divergence… does it make a difference?).

**Weaknesses:**

System does not seem to be generally applicable. It focuses on analyzing repetitive behavioral patterns and general, global, behavioral-influencing agent properties. It seems to me that the tasks indeed are a perfect fit for the presented system.

In the mouse task the distance prediction loss comes out of the blue. An ablation of this one should be conducted. Also the minimal statistics mean, max, min come out of the blue. What is their impact here?

The short- and long-term horizons are fixed and chosen by the system designer. I wonder how robust the system is when varying the two parameters in a reasonable range.

The system assumes separate patterns within the short-term and long-term behavioral dynamics and  the benchmarks offer exactly this.

The authors talk about “realistic robot behavior” – I would remove “realistic” here.

**Questions:**

I had a hard time to understand the multi-scale bootstrapping methodology. I am not familiar with the term “bootstrapping” in this respect – please specify more accurately how it is implemented. It is also a pitty that the two L_r losses are not visualized in Figure 1. What does “positive views” mean here? How are the predictors q_s and q_l implemented.

Can you run further ablations to prove the robustness of your approach?

For which tasks is your approach not suited?

**Limitations:**

Limitations are rather brief and do not really discuss the inducative biases / data assumptions implemented in the system. In fact, the only limitation that is mentioned is the lack of further latent state disentanglement, which I am not quite sure I understand the point here – you very clearly disentangle short- from longer-term behavioral pattern prediction (or rather behavioral histogram prediction). I would rather mentioned the other limitations, which I have mentioned above – including crucial system design choices and critical system parameters.

Please correct the references to tables.

---

> ### Author Rebuttal · Authors · 2023-08-10
>
> Thank you for your comments and questions! We provide point-by-point responses below.
>
> > 1. System does not seem to be generally applicable. It focuses on analyzing repetitive behavioral patterns and general, global, behavioral-influencing agent properties. It seems to me that the tasks indeed are a perfect fit for the presented system.
>
> **Reply:** Thank you for the positive feedback that the tasks that we selected are a perfect fit. While the focus of our current work has been on animal behavior, we do believe that the histogram of actions loss and some aspects of our multi-scale loss should be more generally applicable. We hope to explore more applications of HoA in general time-series analysis in the future.
>
> > 2. In the mouse task the distance prediction loss comes out of the blue. An ablation of this one should be conducted. Also the minimal statistics mean, max, min come out of the blue. What is their impact here?
>
> **Reply:** Thanks for your questions. We agree that the introduction of the mouse distance prediction task and pooling could be better explained. We will revise the text to discuss the rationale and provide better context in a revised version.
>
> To address your question about the impact of these different components, we conducted a number of ablations, which we report in Table 1 of the accompanying pdf. In our first experiment, we removed the interaction loss from the model, meaning that the dynamics of each mouse are modeled completely independently from each other. The ablated model sees a small drop in performance, but continues to outperform all other methods on average. We also ablate the different pooling functions and replace them with a single average pooling operation, where again we find that the drop in performance is minimal. Note that the motivation behind the use of average pooling is to get the average embedding of the mouse triplet, while the max and min help estimate the spread of the embeddings, capturing the difference between their states. We plan to include these ablations in a revised version. Thanks for your suggestions!
>
> > 3. The authors talk about “realistic robot behavior” – I would remove “realistic” here.
>
> **Reply:** We can make that change.
>
>  > 4. “I had a hard time understanding the multi-scale bootstrapping methodology. I am not familiar with the term “bootstrapping” in this respect – please specify more accurately how it is implemented. It is also a pity that the two L_r losses are not visualized in Figure 1.”
>
> **Reply:** The idea of bootstrapping for contrastive learning was originally introduced in Bootstrap Your Own Latent (BYOL) [1]. BYOL learns an embedding of samples (images, in the original paper) that map augmentations of the same sample to similar points in the latent space, and thus needs to start with the notion of “positive views” or a definition of which points should be close in the latent space. In the application of BYOL to temporal data, a positive range of samples or window in time is defined over which all the nearby points in time are considered positive samples. In our model, we have two different windows over which we define positive samples: in the short term space, points within a short distance are mapped to nearby points in the latent space (according to Equation 3), and in the long-term space, all points that are within a larger window get mapped close to each other in their latent space. We will provide more context and details on the loss and related background on bootstrapping in a revised version.
>
> > 5. What does “positive views” mean here? How are the predictors q_s and q_l implemented?
>
> **Reply:** As described in the last response, positive views are defined based upon whether two points in time are within L samples from one another. Views are selected in this window uniformly at random during training. The predictors are small MLPs that aim to predict the representation of one view (called target view) from the other (called online view), we encourage the reviewer to refer to [1]. Both timescales have a separate projector and separate set of latents to build contrast over. We will add this discussion and details on the projectors in the revised paper.
>
> > 6. For which tasks is your approach not suited?
>
> **Reply:** BAMS is designed to build representations that can both capture frame-level dynamics as well as global trends in behavior that occur throughout a sequence. Thus, BAMS is not well suited to work with very short sequences that don’t provide sufficient long-term information. We will provide discussion on limitations and situations in which BAMS may not work well in the discussion section of the revised work.
>
> ---
>
> [1] Grill, Jean-Bastien, et al. "Bootstrap your own latent-a new approach to self-supervised learning." NeurIPS 2023

---

> > ### Comment · Reviewer_hofo · 2023-08-11
> > **Thank you and confirm**
> >
> > Thank you for the replies and the information about the bootstrapping here, which is indeed rather simple (but simple can be good of course).
> >
> > The ablation nicely adds to the paper and underlines the general quality of the approach.
> >
> > As also other reviewers' point out, though, the novelty / impact of the work is somewhat limited and, as I indicated, the benchmarks are perfectly suited for the system, that is, for the algorithm; on the one hand side positively highlighting the system's abilities, but, on the other hand, not really evaluating the scalability / generality of the approach.
> > This should be discussed in detail in the revision, as promised by the authors.
> >
> > Due to the uncertainty about generalizability of the approach and the other limitations I have mentioned I tend to leave my evaluation as is.

---

> > > ### Author Response · Authors · 2023-08-19
> > >
> > > Thanks so much for your reply and engaging in the discussion! We believe the paper is much stronger after the review and we are eager to incorporate the experiments, ablations, and points of discussion suggested by you and the other reviewers into a revised version of the paper.
> > >
> > > Regarding your comments about generalizability of our method, we would like to point you to our latest response to reviewer Uc6v, who requested additional results on the MABe flies dataset. We are excited to report that we achieve state-of-the-art performance on this dataset! The Flies dataset is larger than the mouse datasets, and is evaluated on 50 frame-level and sequence-level tasks; Additionally, the videos are recorded at 150 frames per second, and the number of flies in a sequence can vary from 9 to 11, as opposed to only 3 in the mice data. We wanted to point you to this additional result because we believe it further demonstrates the generalization of our approach to new datasets and scaling to an even larger numbers of animals.

---

> > > > ### Comment · Reviewer_hofo · 2023-08-20
> > > > **nice addition**
> > > >
> > > > good to see another data set tested - in the light of this and the other replies I increase my score not to accept

---

### Official Review · Reviewer_6S6e · 2023-07-12

**Soundness:** 4 excellent
**Presentation:** 3 good
**Contribution:** 3 good
**Rating:** 7
**Confidence:** 3

**Summary:**

The authors propose a multi-task representation learning model for animal behavior consisting of an action-prediction objective and a multi-scale architecture with separate short and long term dynamics. The authors use this model to achieve state of the art performance on the MABe2022 Challenge. Furthermore, they generate a dataset of varying maps for quadruped robots and use the proposed method for behavior predictions in these environments.

**Strengths:**

The paper is well-written with a clear motivation, organization, discussion of mathematical details and experimental results. Diagrams in the method section make the approach particularly clear.

Interesting and novel methodological design with the distinct latent spaces for short and long term dynamics as well as the histogram of actions objective. Also, these methodological components are methodologically simple and increase model interpretability.

State of the art performance on MABe2022 challenge for sequence level tasks and some frame level tasks.

Connection between animal behavior modeling solution and robotics application.

New robot learning dataset of quadrupeds leveraging behavior model for which tasks will be released as a community benchmark.

Code and implementation details are released, making the results easily reproducible.


**Weaknesses:**

While the connection between mouse and robot behavior is interesting, the connection should be made a bit more strongly. What would these behavior models do to aid robotics? This question should be explicitly addressed in the text.

There is a distinct pattern in the results where the frame level results are less competitive than the sequence level. An appropriate discussion and explanation in the text should be added for this performance difference.


**Questions:**

To my understanding, the MABe challenge was held during a fixed timeline, and there may be many papers on this topic which do not treat the dataset as a benchmark. Could you explain if you think this is true, and could you also explain regarding the challenge time?

**Limitations:**

Limitations are discussed, but broader impacts are not. An insight to the authors view of broader impacts of connecting animal behavior models and robotics as they do in their work would be interesting to hear. Also, additional details regarding limitations should be discussed if there are specific robot morphologies you expect this approach to not work well for.

---

> ### Author Rebuttal · Authors · 2023-08-10
>
> Thank you for your comments and questions! We provide point-by-point responses below.
>
> > 1. While the connection between mouse and robot behavior is interesting, the connection should be made a bit more strongly. What would these behavior models do to aid robotics? This question should be explicitly addressed in the text.
>
> **Reply:** In the realm of robotics, the approach for learning robot policies (controllers) from human or animal demonstrations is gaining traction. For instance, to learn complex tasks such as object manipulation or navigation, the data is often gathered by various human tele-operators [1]. Similarly, the animal behavior data can be leveraged as a reference for learning agile and more natural robot locomotion capabilities [2]. Understanding the unique tele-operator or animal traits, biases, and tendencies can help eliminate or harness them for improved robot learning. We plan to add a discussion on these connections in the Discussion section of the paper.
>
> [1] https://learning-from-play.github.io/
>
> [2] https://xbpeng.github.io/projects/Robotic_Imitation/index.html
>
>
> > 2. There is a distinct pattern in the results where the frame level results are less competitive than the sequence level. An appropriate discussion and explanation in the text should be added for this performance difference.
>
> **Reply:** Thanks for the suggestion. Many of the frame-level behaviors in the Mabe benchmark are social behaviors that involve pairs of mice. Because we don’t explicitly model features related to these social behaviors like other approaches (T-PointNet, T-BERT), BAMS doesn’t achieve the state-of-the-art but is still very competitive. At the same time, because we build representations of the long-term and sequence-level dynamics in a separate latent space, our approach really shines in many of the global tasks. We plan to add more discussion of this point in the revised paper.
>
> > 3. The MABe challenge was held during a fixed timeline, and there may be many papers on this topic which do not treat the dataset as a benchmark. Could you explain if you think this is true, and could you also explain regarding the challenge time?
>
> **Reply:** We believe that the most relevant works on animal behavior as well as self-supervised learning methods for sequential data, have been either evaluated by the original authors (like T-VAE [2]) or contributed by the community as part of the challenge (like T-BERT [1]). An extensive benchmark has been released alongside the MABe datasets [1], and we compare against all of the included methods. We benchmark additional self-supervised learning methods like TS2Vec [3], and we have added BYOL [4] in response to reviewer Uc6V. We do not think that the challenge window is a concern, we believe that the benchmark itself is more important than the challenge, because it presents challenging and well rounded tasks for evaluating animal behavior representation methods, which have otherwise been limited to testing on in-house datasets, with limited comparisons against other work.
>
> > 4. An insight to the authors view of broader impacts of connecting animal behavior models and robotics as they do in their work would be interesting to hear.
>
> **Reply:** As we advance towards more fine-grained and naturalistic long-term recordings of behavior in animals and humans, analyzing fine-grained behavior can enhance both robots and human-robot interfaces. In the medical field, these models could be used to develop a more naturally moving robotic controller for prosthetics or exoskeletons. The universality of the controller can be achieved from better understanding of long & short term subtleties of human behaviors, as well as rapidly adapted for the comfort and preference of the final user. Unsupervised data driven approaches can also be powerful for detecting and diagnosing the abnormalities of the complex robot/agent behaviors.
>
> > 5. Additional details regarding limitations should be discussed if there are specific robot morphologies you expect this approach to not work well for.
>
> **Reply:** Currently, our investigations only focus on joint-based robots, and do not have inherent limitations for more complex joint-based morphology designs. There are subfields of robotics that we have yet to explore such as wheeled, buoyancy or soft robotics. While there are no theoretical reasons to believe our approach wouldn’t suit, an investigation and empirical evaluation is needed. Furthermore, our robot experiments currently solely rely on proprioceptive information (robot’s perception of its own body) to extract the behavior models. The integration of exteroceptive sensing will expose the model to a richer information space, which could enable the model to further improve the behavior modeling by observing connections between the robots and its surroundings.
>
> ---
>
> [1] Sun, Jennifer J., et al. "MABe22: A Multi-Species Multi-Task Benchmark for Learned Representations of Behavior." ICML 2023.
>
> [2] Sun, Jennifer J., et al. "Task programming: Learning data efficient behavior representations." CVPR 2021.
>
> [3] Yue, Zhihan, et al. "Ts2vec: Towards universal representation of time series." AAAI 2022
>
> [4] Grill, Jean-Bastien, et al. "Bootstrap your own latent-a new approach to self-supervised learning." NeurIPS 2023

---

> > ### Comment · Reviewer_6S6e · 2023-08-11
> > **Reply to Rebuttal**
> >
> > Thank you very much for your response. I will keep my score.
> >
> > As an additional suggestion while you iterate on paper revisions, from reading the other reviews, I would continue considering how to draw emphasis of this paper's contribution to connecting animal behavior analysis to robotic behavior design and analysis. Phrasing in the introduction and abstract could include minor adjustments to more strongly emphasize this driving goal.
> >
> > For example, using only the mouse component of the MABE dataset seemed well justified to me since you explicitly demonstrate the behavior analysis of your method for quadruped robots and four legged animals. The analysis of the fly component of the dataset would require making a similar connection to quadcopters, a large task beyond the scope of this work in my view. Analyzing general animal behavior outside the connection with robotic behavior design seemed separate from the primary intent of this work to me.

---

> > > ### Author Response · Authors · 2023-08-19
> > >
> > > Thanks for your continued positive view of our work. We appreciate your feedback and suggestions, and will definitely further highlight this connection in our revision.

---

### Author Rebuttal · Authors · 2023-08-10

We would like to thank all of the reviewers for their thoughtful feedback and helpful suggestions! We are delighted to hear that all of the reviewers appreciated the work and motivation, and were impressed with the fact that BAMS achieves state-of-the-art performance on a complex realworld multitask benchmark.

Strengths highlighted by the reviewers:
- **Innovations of new distributional loss for forward prediction:**  (7X95) “an interesting innovative approach by utilizing histograms for predictions” (hofo) “An innovative histogram-based loss component”
Method and approach:  (6S6e) “Interesting and novel methodological design with the distinct latent spaces for short and long term dynamics as well as the histogram of actions objective.” (hofo) “learns to characterize behavioral patterns, solely by post-training a linear classifier given the emergent latent codes” (7X95) “very relevant to the research community” (Uc6v) “Ablation analysis is good, and gives insights into design choices.”
- **Writing and presentation:** (7X95) “The paper is well-written and maintains a clear and organized structure, ensuring a pleasant reading experience.”  (Uc6v) “Paper well written and easy to follow.” (6S6e) “The paper is well-written with a clear motivation, [...] Diagrams in the method section make the approach particularly clear.”
- **State-of-the-art performance on a challenging real world benchmark:**  (6S6e) “State of the art performance on MABe2022 challenge” (7X95) “the paper highlights the effectiveness and potential practical application of the approach.”

**Main revisions and new experiments:**

To address reviewer suggestions and concerns, we ran three sets of experiments:
- A new BYOL baseline to address Reviewer Uc6V’s comment regarding the inclusion of additional contrastive learning baselines for Table 2,
- Ablations of the multi-mouse interaction loss and pooling operations as suggested by Uc6V and hofo, to address questions about the impact of these components on final performance,

---

### Decision · Program_Chairs · 2023-09-21

**Decision:**

Accept (spotlight)

**Comment:**

This paper present a new approach to multi-scale modeling of dynamical systems (i.e., time-series).
This is a very important topic, and this contribution is timely and likely to be of interest to a wide audience.

All reviewers agree that the approach proposed in novel and interesting.
The experimental results are thorough and include a diverse set of datasets.